# Geometry-Aware Texture Generation for 3D Head Modeling with Artist-driven Control

## Abstract

*Creating realistic 3D head assets for virtual characters that match a precise artistic vision remains labor-intensive. We present a novel framework that streamlines this process by providing artists with intuitive control over generated 3D heads. Our approach uses a geometry-aware texture synthesis pipeline that learns correlations between head geometry and skin texture maps across different demographics. The framework offers three levels of artistic control: manipulation of overall head geometry, adjustment of skin tone while preserving facial characteristics, and fine-grained editing of details such as wrinkles or facial hair. Our pipeline allows artists to make edits to a single texture map using familiar tools, with our system automatically propagating these changes coherently across the remaining texture maps needed for realistic rendering. Experiments demonstrate that our method produces diverse results with clean geometries. We showcase practical applications focusing on intuitive control for artists, including skin tone adjustments and simplified editing workflows for adding age-related details or removing unwanted features from scanned models. This integrated approach aims to streamline the artistic workflow in virtual character creation.*

## 1. Introduction

Creating immersive virtual experiences relies on realistic character heads with diverse appearances across video games, films, and virtual reality applications. This requires generating both detailed geometry and high-quality texture maps for convincing rendering. Traditional approaches include capturing real subjects in light stages [7] or manually sculpting in specialized software like ZBrush[1], but these methods remain labor-intensive, requiring significant time investment and specialized skills from artists.

Alternative approaches to 3D head generation rely on 3D Morphable Models (3DMMs) [4], which create statistical models from head scan collections [10]. While these linear models offer convenient generation through basis combination, they typically produce overly smooth results lacking the fine details necessary for realism. More recent nonlinear approaches using Generative Adversarial Networks [15] can generate more detailed heads [26, 37] with corresponding texture maps. However, these methods often require additional post-processing steps to achieve satisfactory results.

Other generative based methods, whether image-conditioned [24, 25, 35] or text-prompted [48, 53], offer limited artistic control. They typically only allow global modifications rather than precisely placed details like scars or wrinkles, and often require multiple iterations to achieve a specific artistic vision, particularly when trying to realize precise facial attributes or skin tones through text descriptions.

We present a novel framework that provides artists with intuitive control at three levels while streamlining the creation process. First, our approach generates high-quality head geometry with corresponding textures that maintain consistency across different demographics. Second, we enable precise skin tone manipulation without altering other facial characteristics, which helps both artistic expression and representation of diverse populations. Third, we facilitate detailed editing of specific features like wrinkles or facial hair. All of these controls are integrated into a cohesive system that automatically propagates edits across all necessary texture maps. Our contributions include:

- A multi-level control framework for 3D head generation that enables artists to manipulate geometry, skin tone, and fine details independently.
- A geometry-aware texture generation approach that maintains consistency between facial structure and appearance.
- A novel approach for editing fine-grained details in intrinsic texture maps. This allows artists to modify a single texture map using any image editing tool, with the changes coherently propagated to the intrinsic texture

---

[1]https://www.maxon.net/en/zbrush

maps.

- A data-driven method for precise manipulation of skin tones while preserving other facial attributes, enabling greater diversity in virtual character creation.

## 2. Related works

**Generative head models** The generation of faces and full heads has traditionally relied on statistical morphable models (3DMM) [4, 5, 13, 27]. These linear models are constructed by applying Principal Component Analysis (PCA) to a collection of head scans. The resulting orthogonal basis enables the sampling of new heads through linear combinations of basis vectors. However, due to the lack of semantic interpretation in this basis, controlling or editing the generation process remains unintuitive. Moreover, the generated heads lack fine details, as high-frequency information is not captured by PCA. Non-linear morphable models [12, 26, 37] have been shown to produce more detailed heads compared to linear models [12]. These non-linear models can be categorized into two main types: the first leverages Convolutional Neural Networks (CNNs) and Generative Adversarial Networks (GANs) [15] operating in UV space [26], while the second utilizes Graph Neural Networks (GNNs) that operate directly on vertex coordinates [2].

The approaches using CNNs consider the generation of geometry and color within the 2D UV space as in [26]. This requires flattening the 3D geometry into UV space. However, flattening the geometry onto a 2D space without making cuts isn't feasible. As a result, some areas that are close together in 3D end up being far in the 2D representation. Conversely, some positions that are close on the 2D UV map can be far apart in the 3D mesh, such as the eyes and mouth. This creates problems for using CNNs in UV space, as it violates the inductive bias of locality in CNNs. This leads to visible artifacts on the geometry. Post-processing is generally applied to fix these artifacts as in [26].

The second approach for generative head models uses Graph Neural network (GNN) and auto-encoders to generate a complete head without requiring additional post-processing [2, 38]. However, as they operate on a vertex-level basis, they are unable to generate sharp and detailed color texture maps.

In this work, we propose a geometry-aware texture synthesis network, where the geometry is driven by a GNN-based auto-encoder, and the color texture generation is based on CNNs, for a complete and high-quality head generation.

**Controllable generative models** Controlling generative models has achieved remarkable advances for 2D portrait image editing, mainly because of the availability of a large corpus of 2D portrait image datasets such as [21, 29]. Such methods [1, 18, 28, 33, 40, 41, 51, 52, 54], allow for semantic editing of the face attributes on 2D portrait images. This is achieved by projecting the image onto the latent space of StyleGAN [22] and finding orthogonal directions in that space that allow for controlling various face attributes (such as hair, age, expression, and eyeglasses). Recently, [45] uses a pre-trained StyleGAN [23] to control the skin tone of a 2D portrait image.

Although numerous works tackled the problem of controlling generative models for 2D portrait images, there is a scarcity of work addressing this problem for 3D facial asset generation. When referring to 3D facial assets, we consider both the geometry (coordinates of vertices in 3D space) and the associated texture maps (including diffuse, specular, and normal maps) used for realistic rendering. StyleRig [44] drives a pre-trained GAN network with 3D morphable model (3DMM) [4]. Following the same direction, Albedo-GAN [37] uses a pre-trained StyleGAN to generate a 2D image and then recover back the 3DMM parameters. These models allow controlling the expression, head pose, identity, and scene lighting of the generated image. However, they inherit the limitations of 3DMMs and do not provide artists with precise control. For instance, artists manipulate and edit directly the texture maps and the geometry to achieve the desired output. Other methods allow for a specific control, such as gender and age [26] or including ethnicity and body mass [12]. In [31, 32] the artists control the texture generation model using a segmented feature map to specify colors and specify global attributes through tags. More recently, diffusion-based models [48, 53] were proposed to produce realistic head avatars from text prompts. However, these methods lack fine-grained control over the generation process making them less usable for artists.

In contrast to previous work, our approach offers fine-grained artistic control over a generative model, tailored to fit the artists' workflow. This is achieved by designing a pipeline that takes explicit artist control at intermediate points of the generation process, enabling them to interact with generated assets by independently adjusting skin tone and fine-grained geometric details.

**Skin tone color control** Precise adjustment of the skin color of virtual characters is crucial for artists when designing worlds with specific intent. Furthermore, tweaking skin tone can prove valuable in addressing biases towards underrepresented ethnicities. Moreover, this capability enables the creation of diversity and enhances the realism of the user experience. However, manipulating skin tone on a large scale can be challenging and time-consuming.

The skin color of human skin is determined by the levels of melanin and hemoglobin concentration in the epidermis layer [3, 16]. This information serves as a basis for modifying the skin tone of virtual characters. However,

Figure 1. Overview of the proposed pipeline. Generator $\mathcal{F}$ generates a mesh and $G$ generates an intermediate skin representation: a melanin map $\mathcal{M}$ and a high-frequency details map $\mathcal{H}$. $G_{\mathcal{A}}$ generates a color map $\mathcal{A}$ from $\mathcal{M}$, allowing for precise control of skin tone. $G_{\mathcal{C}}$ generates a final skin reflectance map $\mathcal{C}$, incorporating arbitrary manipulations on $\mathcal{H}$. $G_{\epsilon}$ decomposes the intrinsic face reflectance maps, which can be used to render realistic heads.

accurately capturing melanin and hemoglobin concentrations is a resource-intensive task that requires specialized equipment and procedures [14, 34]). Some work aims to estimate these properties from images, [46, 47] use independent component analysis (ICA) to estimate melanin and hemoglobin distributions in 2D portrait face images. Donner *et al.* [9] proposed a parametric skin reflectance model based on melanin and hemoglobin concentrations, enabling control over skin color. Nevertheless, understanding the complex relationship between melanin/hemoglobin concentrations and skin color is challenging, with limited literature available on obtaining an explicit relationship between skin color and the melanin-hemoglobin representation. In this work, we introduce a data-driven approach that learns the implicit relationship between the melanin-hemoglobin space and the color space.

## 3. Method

Our head generation pipeline provides multiple levels of artistic control by formulating sub-tasks that directly accept the user input. Figure 1 illustrates our pipeline. The geometry generator $\mathcal{F}$ (Section 3.1) produces a mesh from a latent code $\mathbf{z}_g$. The texture generator $G$ (Section 3.2) conditioned on $\mathbf{z}_g$ outputs two intermediate texture maps: the skin tone control map $\mathcal{M}$ and high-frequency details map $\mathcal{H}$). These two maps establish distinct control paths for artists, enabling them to separately manipulate skin tone and fine-grained skin details. Model $G_{\mathcal{A}}$ (Section 3.3) enables skin tone control with a single scalar $\alpha$. The high-frequency map $\mathcal{H}$ can be freely modified by an artist, to add or remove fine-grained details. Model $G_{\mathcal{C}}$ (Section 3.4) processes the concatenation of $\mathcal{H}$ and $\mathcal{A}$ (which modification can be done on both of these texture maps) producing the skin reflectance map $\mathcal{C}$. These modifications are then propagated to the intrinsic texture maps via model $G_{\epsilon}$ (Section 3.5). Finally,

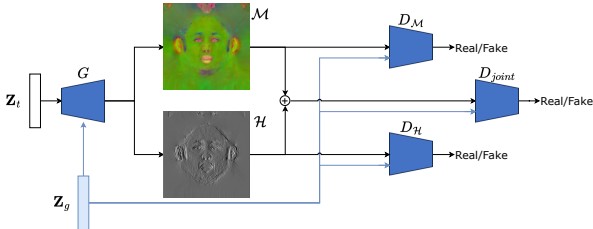

Figure 2. Geometry-aware texture generation. Given $\mathbf{z}_g$ that defines a head's geometry, $G$ estimates skin tone control maps $\mathcal{M}$ and high-frequency skin details $\mathcal{H}$.

super-resolution is applied to generate 4K textures for rendering.

### 3.1. Geometry generation

For geometry generation, we train a part-based Variational Autoencoder (VAE) similar to [2]. This VAE consists of eight Graph Neural Network (GNN) encoders [38], each encoding the vertices of a specific facial region into a latent representation. These representations are concatenated into a single vector, $\mathbf{z}_g$, which is then passed to a decoder, $\mathcal{F}(\mathbf{z}_g)$, to reconstruct the full input mesh. We adopt the same network architecture as in [2].

To generate new samples for different classes (e.g., gender, age, or ethnicity), we follow these steps: For a given class $c$, we calculate the mean $\boldsymbol{\mu}_c$ and standard deviation $\boldsymbol{\sigma}_c$ of the latent codes within that class. We sample new latent codes $\mathbf{z}_g \sim \mathcal{N}(\boldsymbol{\mu}_c, \boldsymbol{\sigma}_c^2)$ to generate samples belonging to a specific class, allowing artists to specify the desired class for geometry generation. Unconditioned samples can also be generated using $\mathbf{z}_g \sim \mathcal{N}(\mathbf{0}, I)$.

## 3.2. Geometry-aware texture generation

We train a geometry-aware generator, $G$, that produces two intermediate representations for skin textures: skin tone control maps $\mathcal{M}$ and high-frequency details $\mathcal{H}$, as illustrated in Figure 2. The skin tone control map $\mathcal{M}$ is a three-channel image encoding melanin and hemoglobin features, forming the foundation for skin tone generation (Section 3.3). The high-frequency details map $\mathcal{H}$, represented as a single-channel texture map, captures facial details such as wrinkles and scars. The intuition behind this separation is to offer an artist separate control over the skin color and high-frequency details.

To encourage the network to learn the correlation between head geometry and textures (e.g. for different ethnicities or perceived gender), we condition the texture generator on the latent code $\mathbf{z}_g$ of the mesh geometry. More precisely, the geometry latent code $\mathbf{z}_g$ is concatenated with a random noise vector $\mathbf{z}_t \sim \mathcal{N}(\mathbf{0}, \mathbf{I})$, allowing the generation of multiple textures for the same geometry.

During training, we employ three distinct discriminators: (i) one for the skin tone control map $\mathcal{M}$, (ii) one for the high-frequency details map $\mathcal{H}$, and (iii) one for the combined representation of $\mathcal{M}$ and $\mathcal{H}$ to learn their correlation. Additionally, these discriminators incorporate the latent code $\mathbf{z}_g$ as a conditioning input.

## 3.3. Skin tone control

The skin tone control network $G_{\mathcal{A}}$ maps the skin tone control map $\mathcal{M}$ and a scalar $\alpha$ representing melanin concentration power to the corresponding color texture map $\mathcal{A}$. This enables precise skin tone control using a single scalar $\alpha$.

Establishing an analytical and physical relationship between the melanin-hemoglobin space and the color space is a non-trivial challenge. Therefore, we adopt a data-driven approach to learn this mapping. To achieve this, we construct an artist-curated dataset consisting of tuples $\mathcal{M}, \alpha, \mathcal{A}$, where $\alpha$ represents melanin concentration. Lower $\alpha$ values correspond to lighter skin tones, while higher values produce darker skin tones. The texture map $\mathcal{A}$ is a smooth representation that removes high-frequency details while preserving the fundamental skin tone of the reflectance map. Further details on $\mathcal{A}$ are provided in Section 4.1. To learn the relationship between $\mathcal{M}$ and $\mathcal{A}$, we train an image-to-image translation network [19] with U-Net architecture [39], denoted $G_{\mathcal{A}}$. We use the multi-patch, multi-resolution discriminator proposed in [36]. This network takes $\mathcal{M}$ as input, along with an additional channel where all positions contain the scalar $\alpha$. During training, we minimize both the adversarial loss and the $\ell_2$ distance between the network's output and the corresponding ground-truth texture map.

Once trained, $G_{\mathcal{A}}$ effectively maps the melanin-hemoglobin space to the skin color space. During inference, given a melanin-hemoglobin map (generated by $G$), we can dynamically adjust the skin tone by varying the input $\alpha$.

## 3.4. Fine-grained detail editing

The final reflectance map $\mathcal{C}$ is obtained by the network $G_{\mathcal{C}}$ that uses low-frequency details (skin color map $\mathcal{A}$) and high-frequency details map $\mathcal{H}$. Formulating the generation process with these intermediate steps allows convenient manipulation of the high-frequency details of the face. An artist can make arbitrary changes to $\mathcal{H}$ using image-editing software to add/remove small details.

To train this network, we construct a dataset of tuples containing reflectance maps $\mathcal{C}$ along with their corresponding high-frequency and low-frequency components. The high-frequency map $\mathcal{H}$ is extracted by applying a Sobel filter [42] to $\mathcal{C}$.

For training, we use an image-to-image translation network [19] with a U-Net architecture [39], denoted as $G_{\mathcal{C}}$. Following [19], this network takes as input the concatenation of the skin color map $\mathcal{A}$ and the high-frequency map $\mathcal{H}$ and learns to reconstruct the final skin reflectance map $\mathcal{C}$. To enhance output quality, we employ the multi-resolution discriminator from [36]. Training involves minimizing both the adversarial loss and the $\ell_2$ distance between the network's output and the corresponding ground-truth texture map.

## 3.5. Face attributes decomposition

The intrinsic face attributes (specular $\mathcal{S}$ and normal $\mathcal{N}$ map) are estimated using two separate networks: $G_{\mathcal{E}_s}$, and $G_{\mathcal{E}_n}$, respectively. The role of these networks to ensure that the modifications made by the artist are propagated into skin reflectance maps. These maps are essential for realistic rendering, especially under novel lighting conditions. For $G_{\mathcal{E}_s}$, we use a translation network [19] with U-Net architecture [39] with a multi-patch, multi-resolution discriminator from [36]. For $G_{\mathcal{E}_n}$, we follow the same approach as in [8, 26, 50], by first predicting the displacement map using a patch-based approach. The normal map is then obtained from the displacement map using a Fast Fourier convolution.

The generated texture maps are then upsampled using a pre-trained super-resolution network [49] fine-tuned on our dataset. Similar to [26], the upscaling process is performed in two stages: initially, the feature maps are upsampled to $1024 \times 1024$, then to 4k ($4096 \times 4096$ pixels).

## 4. Experimental Protocol

## 4.1. Dataset

We use a dataset consisting of 892 head scans captured using a light stage. Figure 5 shows one sample from the dataset. For each subject in the dataset, we have the face geometry (vertices positions $V$), the skin reflectance map $\mathcal{C}$

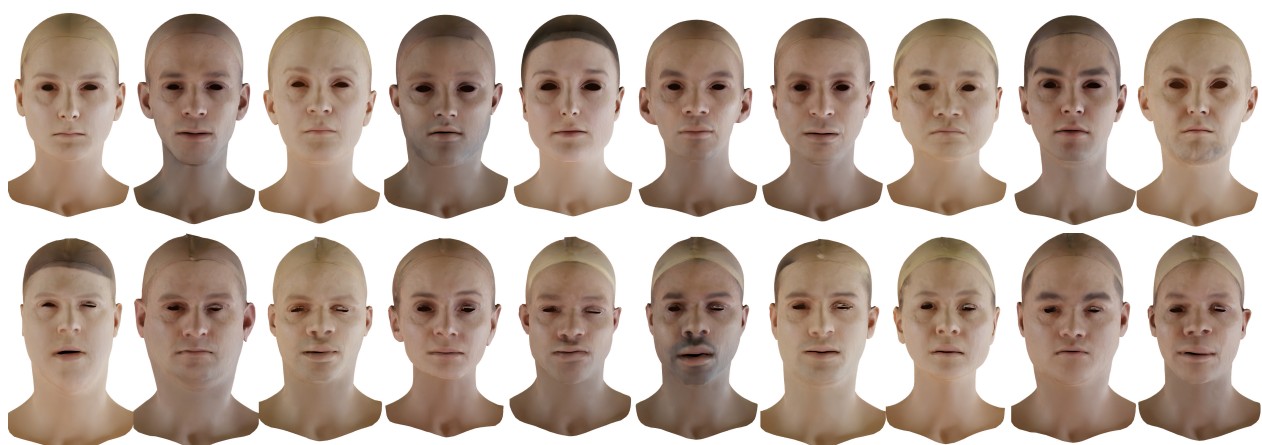

Figure 3. Top: Samples from our model. Bottom: Samples from the baseline

(different resolutions from 256 to 4096), the specular $\mathcal{S}$ and normal map $\mathcal{N}$.

To obtain the skin color map $\mathcal{A}$, we apply a Principal Component Analysis (PCA) over the reflectance map $\mathcal{C}$ for the entire training dataset, retaining only the first 15 eigenvectors. Subsequently, we project each $\mathcal{C}$ onto the PCA basis. The result is a low-frequency image with the base skin tone of each subject. We obtain $\mathcal{H}$ using the Sobel operator [20], which captures high-frequency details such as folds, wrinkles, moles, and pores. We use a curated dataset of maps $\mathcal{M}$ for melanin-hemoglobin representation, based on results from a commercial tool[2].

### 4.2. Implementation

The geometry decoder $\mathcal{F}$ uses the same architecture and training procedure as described in [2]. For the texture generator $G$, we follow the training procedure from Style-GAN2 [22], with the change of using one generator and three discriminators (see section 3.2). We train $G$ for 1500 epochs, with a batch size of 8 and a learning rate of 0.002. Both $G_{\mathcal{A}}$ and $G_{\mathcal{C}}$ are trained for 300 epochs with a batch size of 2. The initial learning rate is equal to 0.0001 which we decay by 0.1 every 60 epochs. We train $G$ and $G_{\mathcal{A}}$ using $256 \times 256$ texture resolution. $G_{\mathcal{C}}$ is trained with $512 \times 512$ texture resolution, as it has shown to produce better results. During inference, we upsample $\mathcal{A}$ and $\mathcal{H}$ to $512 \times 512$ to match the resolution of $G_{\mathcal{C}}$. For networks $G_{\mathcal{E}s}$ and $G_{\mathcal{E}n}$, we use the same architecture and training scheme as detailed in [8].

### 5. Results and Discussion

#### 5.1. Geometry and texture synthesis

In this section we evaluate three aspects of geometry and texture synthesis: (i) we compare our GNN generator operating on vertices to a CNN generator operating on UV

[2]https://texturing.xyz/

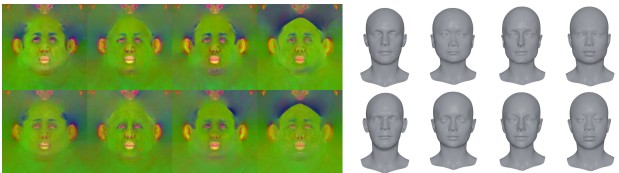

Figure 4. Top: Randomly generated samples from our model. Bottom: Closest sample in the dataset to the generated sample.

space, (ii) we study the impact of conditioning our texture synthesis model with geometry. Finally, (iii) we evaluate the ability of our model to produce novel heads beyond the training data.

To highlight the benefit of using a GNN for geometry generation, we compare our method to a StyleGAN-based generator that jointly generates texture and geometry in UV space, similar to [26]. We refer to this model as the "baseline". We note that [26] estimates only the frontal part of the face with a non-linear model that is later fused with the remaining head parts that are estimated by a separate linear model. Here, we compare non-linear models that estimate the entire head *without* any post-processing steps.

Figure 3 shows randomly sampled heads obtained using both methods. Our method generates more plausible results. In general, faces have significantly fewer artifacts around the eyes and mouths than in the baseline method. The artifacts produced by the baseline stem from points that are neighboring in the UV representation (e.g., the top and bottom of the eyelids), but are disjoint in 3D space. This causes 2D convolutional layers to exploit local 2D neighborhoods that are sometimes nonexistent in 3D. A GNN that operates directly on the vertices of a mesh does not have this problem.

In Table 1 we compare how our method and the baseline generate plausible heads compared to the training set

| Method | FID ↓ |
|---|---|
| Ours | **11.44** |
| Ours un-conditioned | 11.53 |
| Baseline | 21.28 |

Table 1. Quantitative comparison between our method and the baseline.

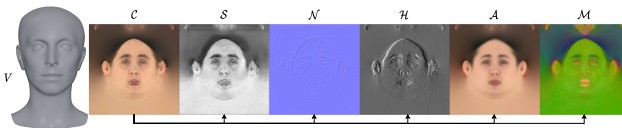

Figure 5. A sample from the training set: mesh vertices $V$, color-map $\mathcal{C}$, specular map $\mathcal{S}$, Normal map $\mathcal{N}$, high-frequency map $\mathcal{H}$, skin-color map $\mathcal{A}$, skin tone control map $\mathcal{M}$.

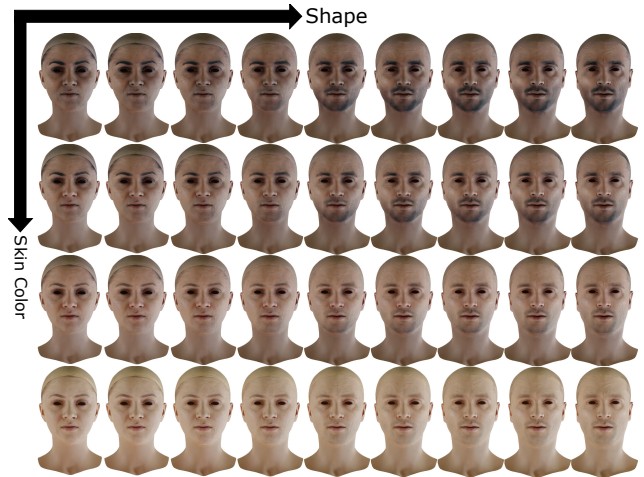

Figure 6. Left to right: shape interpolation. Top to bottom: Skin color control with the melanin power $\alpha$

distribution. We render 10K images from the geometry and textures estimated by each method to calculate the Fréchet Inception Distance (FID) [17] to the ground truth renders. FID compares the distribution of the generated images to a set of real images, using the features of an inception-v3 model [43]. Our GNN-based model performs significantly better than the baseline method, which indicates that our generated heads have a distribution closer to the training data. We also measure the effect of conditioning our texture generation with the latent vector $\mathbf{z}_g$ that describes the geometry. The conditioned version of our models performs slightly better than its unconditioned counterpart. This indicates that conditioning texture generation on geometry leads to more correlated textures and geometry.

Next, we assess the capacity of our generator to produce novel heads and not simply memorize the training data. Figure 4 presents generated samples alongside their closest matches from the dataset, showing that our method produces heads that are visually different from the content of the training dataset.

To quantitatively evaluate the generalization capacity of our generator, we perform the following experiment: We generate a set of 10k heads using our pipeline. For each generated mesh $V$ and skin tone control map $\mathcal{M}$, we find the closest sample in the dataset using the L2 distance and report the average value. As a reference, we also compute this metric with samples on the dataset and their closest matches. Table 2 shows the results. We notice a similar dis-

|  | data-data | generated-data |
|---|---|---|
| L2 distance (Geometry) | 0.38 | 0.35 |
| L2 distance (Melanin-hemoglobin) | 0.43 | 0.45 |

Table 2. The mean L2 distance between closest samples in the dataset (data-data), and between generated samples and their closest match in the data (generated-data).

tance between dataset-dataset closest pairs and generated-dataset closest pairs, showing that the model generates diverse samples.

## 5.2. Skin tone manipulation

Our pipeline enables precise skin tone manipulation with a single scalar: the melanin power $\alpha$ (explained in section 3.3). Figure 6 shows that linear interpolations of $\alpha$ provide plausible skin tone variations for the same generated head. We show that this control of skin tone works for a variety of face shapes. From left to right, figure 6 shows a linear interpolation between two geometry latent codes. We also observe that network $G$ gradually generates the beard as we move from left to right. This demonstrates that $G$ correlates the generated textures with the input mesh.

To quantitatively evaluate the accuracy of our skin tone control, we compare our method to a baseline that consists of manipulating the hue-saturation-value (HSV) of the texture, which is a common approach used by artists for skin color editing. We hypothesize that our method is better at matching the lip color for a given skin color. We designed an experiment to evaluate this, as follows: we generated 1000 heads using our pipeline. For each head, we generate two textures with different $\alpha$: a source $\mathcal{C}_{src}$, and a target $\mathcal{C}_{tgt}$. We use an optimization procedure to find the optimal HSV value that, added to $\mathcal{C}_{src}$, matches the target skin color $\mathcal{C}_{tgt}$. We denote the resulting texture $\mathcal{C}_{hsv}$. Given $\mathcal{C}_{tgt}$ and $\mathcal{C}_{hsv}$, we find the 5 closest textures in the training dataset in terms of skin tone using the Individual typology angle (ITA) distance [6, 11, 30], that measures skin tone with a single scalar. Finally, we compute the average error and standard deviation on the Hue component between the generated textures and their corresponding closest textures in the dataset, on the lips region. Table 3 reports the average

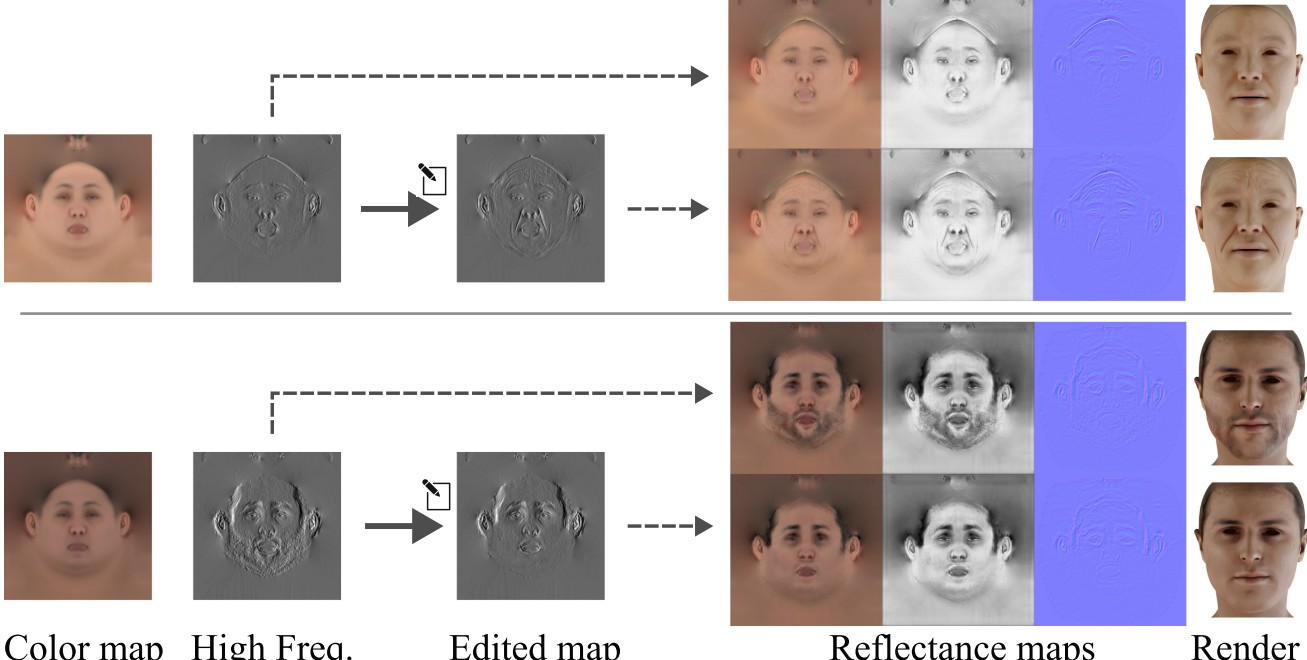

Color map    High Freq.    Edited map    Reflectance maps    Render

Figure 7. Example of artistic editing of high-frequency details. We show an example of adding wrinkles (top subject) and removing a beard (bottom subject). Changes in the single-channel high-frequency map are cohesively propagated to the reflectance maps

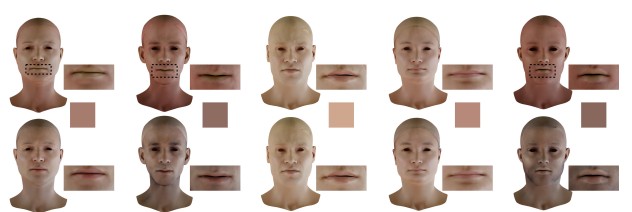

Figure 8. Results on skin tone editing with HSV (top) and Ours (bottom).

| Method | Mean ↓ | Std deviation ↓ |
|--------|--------|-----------------|
| HSV    | 0.295  | 0.235           |
| Ours   | **0.252** | **0.116**    |

Table 3. Error on the lips region of our method compared to HSV skin tone editing.

error and standard deviation on the Hue component for our method and the HSV method. Our method demonstrates significantly lower error compared to the HSV method and exhibits significantly lower variation. This shows that our model is better than the HSV editing at correlating lip color with skin tones. Figure 8 shows renderings of both methods. We noticed that the HSV approach obtains an unrealistic, greenish color for the lips, while our method produces natural lips color matching the face skin tone. Moreover, the HSV-based linear model tends to generate reddish colors for dark skin tones, while our model produces a more realistic rendering.

### 5.3. Fine-grained details manipulation

Our pipeline enables manipulations of fine-grained face details in a single-channel map $\mathcal{H}$ while preserving the skin tone of the subject. Artists can make arbitrary changes on this map (using off-the-shelf editing tools), that are cohesively propagated to the intrinsic facial reflectance maps. To validate the usefulness of this property, we tasked an artist to make modifications for two subjects generated from the model. In the first task, the artist was asked to add wrinkles to make a character look older. In the second task, the artist was asked to remove the beard of a subject - a common task required to clean up scans, as beards are generally modeled separately from the skin in rendering engines. Figure 7 shows the result of this process. We notice that changes to the Fine-grained details map $\mathcal{H}$ are properly propagated to all reflectance maps while preserving the original skin color of the subject, and consequently, simplifying the process of making these changes. Figure 7 shows additional experiments conducted on edge editing.

### 5.4. Limitations

Even if our GNN-based geometry generator produces fewer artifacts than its CNN-based counterparts, on a few subjects, it is possible to see some unnatural wrinkles and folds.

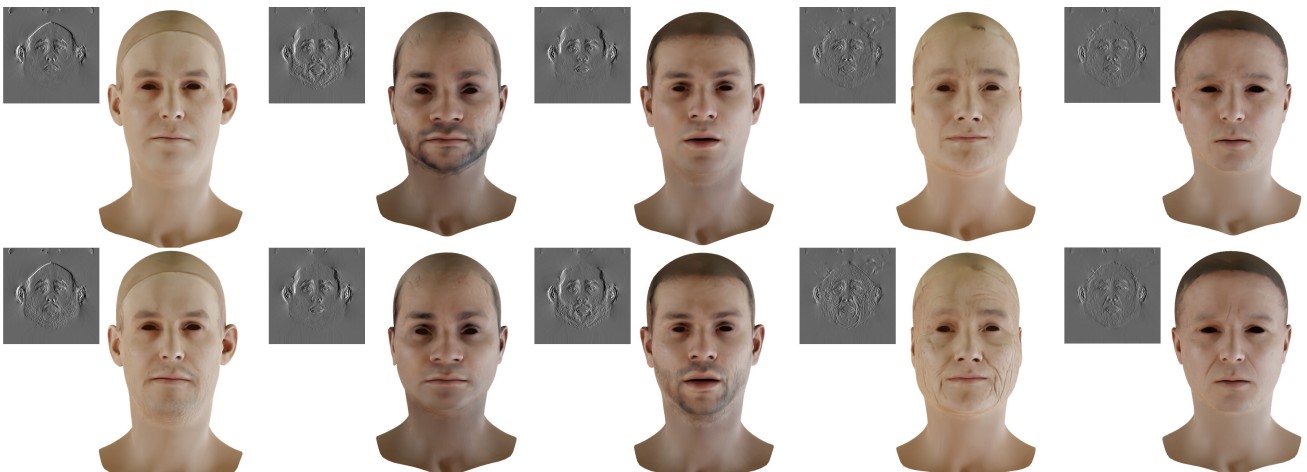

Figure 9. Additional examples of high-frequency detail editing. Top: Render of the original asset. Bottom: Render after changes to the map $\mathcal{H}$.

We intend to investigate new regularization or statistically-based filtering techniques to repair or reduce the occurrence of these problems. In addition, there seems to be implicit factor correlations in our latent space. We intend to explore strategies to enforce disentanglement to further improve controllability.

## 6. Conclusion

We presented a novel framework for creating 3D head assets that offers artists intuitive control at multiple levels. By combining vertex-level geometry generation with a geometry-aware texture synthesis pipeline, our approach produces consistent and diverse results while streamlining the artistic workflow. The three-level control system of geometry manipulation, skin tone adjustment, and fine-grained detail editing provides significant flexibility for realizing specific creative visions.

Our skin tone manipulation method enables precise control while preserving other facial characteristics, addressing both artistic needs and potential dataset biases. The coherent propagation of edits across all texture maps from a single source significantly reduces the time and effort required for common tasks such as adding age-related details or removing unwanted features from scanned models.

Future work could extend this approach to include dynamic facial expressions (e.g. dynamic wrinkle maps). Additionally, expanding the framework to handle full-body avatars with the same level of intuitive control represents another promising direction for research.

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
