# OpenReview forum: "Geometry-Aware Texture Generation for 3D Head Modeling with Artist-driven Control"
_thecvf.com/CVPR/2025/Workshop/CVEU — CVPR 2025_

### Official Review · Reviewer_toAc · 2025-03-13

**Rating:** 4
**Confidence:** 4

**Review:**

Summary

This paper presents a face generation system that both generates geometry and texture and easy-to-edit capability for artists. To this end, two generators, each for texture and geometry, are used. The texture generator generates melanin map and high-frequency details map to make the skin tone edit easy. For the consistency between geometry and texture, the texture generator takes the generated geometry information into account. Following modules output color map, skin reflectance map, specular map, and normal map for the final face renderings.

Strengths

The proposed system is practical and enables easy edits for artists.

Weaknesses
1. No hair and eyes. All the generated heads do not have hair and eyes. This limits the expressiveness of the generated ones quite strictly.

2. The hair colors are related to the skin colors. Fig. 6 shows that as the skin colors become brighter, hair colors also become brighter, which is not desirable.

3. Marginal improvements of the geometry condition in Table 1. The FID difference between the first and second rows are marginal.

4. Writings are not very clear. Top-down approach, which states the conclusion and important things at the first of the paragraph, is recommended.

Questions

1. Which renderer did the authors use? PBR? It is not clear.

---

### Official Review · Reviewer_bz42 · 2025-03-18
**Review:**

**Rating:** 4
**Confidence:** 4

**Review:**

- Clarity: The paper is clear and proposes a 5 part framework for generating controlable geometry, texture, colors, and fine details of a head geometry. A point is that in Section 5.1 the baseline used to compare against the GNN for the mesh generation seems to be a much weaker version than that in [26] (see lines 368-375).  Another point of concern here is not much is shared about the details of the architectures of the models used.
- Novelty: Seems incremental as it combines previously proposed models, however uses a new dataset and have a good motivation for the proposed pipeline and the models chosen for each step in the pipeline.
- Quality and Significance: We believe that despite some drawbacks identified, this work can be useful to the 3d modelling community.

---

### Official Review · Reviewer_T1cb · 2025-03-22
**Promising direction yet suffer from weak novelty and limited evaluation**

**Rating:** 2
**Confidence:** 3

**Review:**

**Strengths**

- This study presents a valuable, timely exploration of controlling 3D head generation.
- The proposed framework covers several important attributes, including head geometry classes, skin tone, and facial details.
- The technical details are easy to follow. The video further provides intuition behind the framework and demonstrates the convenience in manipulating low-level details.


**Weakness**

M1. The "artist control" mechanism is a useful addition but does not constitute a fundamental innovation—many prior works have introduced similar attribute control mechanisms. Additionally, the intermediate representations (e.g., melanin maps and reflectance maps) follow the standard practice of disentangling generative components for interpretability and realism. Without a novel architecture, training paradigm, or significant theoretical insight, the contribution feels incremental rather than groundbreaking.

M2. Lack of evaluation of the coherence when propagating changes. In the claimed contribution to enable control, it remains largely unclear to what extent the proposed pipeline facilitates design iteration with coherent and satisfying change propagation. The authors may include more quantitative evaluations, such as measuring preservation of unedited features

M3. Weak assessment of the novelty in generated heads. Figure 4 does not convincingly demonstrate the capacity of the proposed framework to produce novel heads with merely eight instances, without an overview of the generated samples pool and the original dataset.

M4. Disconnected from the empirical understanding of an artist's workflow. While the authors claimed to tailor the model to fit the artists' workflow (Line 161), there is no evidence or direct justification for designing the pipeline in a way that supports head geometry, skin tone, and facial details in a sequential manner. For instance, I doubt generating the head geometry from a class code can effectively "match a precise artistic vision". The authors may revisit the literature or involve artist evaluators in this aspect.


**Questions**
- Performance-wise, what is the average time cost for a forward pass? How long does it take to train the model?
- Table 2 has a relatively small scale. Was the data normalized and how?

**Minor Issues**
- In Fig. 1, the artist controls are not explained in the caption.

---

### Decision · Program_Chairs · 2025-03-25

**Decision:**

Accept

**Comment:**

Strengths:
The paper provides a timely and practically valuable exploration into controlling 3D head generation, clearly detailing a pipeline that enables artists to manipulate geometry, skin tone, and fine facial details. Its technical descriptions are clear and intuitive, and it leverages intermediate representations effectively, facilitating practical editing workflows beneficial to the 3D modeling community.

Weaknesses:
The contribution is incremental, lacking significant architectural innovation or theoretical insights beyond existing attribute-control approaches. Evaluations lack sufficient quantitative assessment of coherence and generative novelty, baselines appear weakly justified, and critical expressive elements (such as hair and eyes) are missing. Additionally, clarity issues in the presentation and insufficient alignment with real artist workflows diminish the paper's overall impact.

Given the mixed scores, we have decided to accept the paper. The authors should prepare the camera-ready version by carefully addressing the weaknesses identified by the reviewers.